# Regeneration of Hair Cells from Endogenous Otic Progenitors in the Adult Mammalian Cochlea: Understanding Its Origins and Future Directions

**DOI:** 10.3390/ijms24097840

**Published:** 2023-04-25

**Authors:** Natalia Smith-Cortinez, A. Katherine Tan, Robert J. Stokroos, Huib Versnel, Louise V. Straatman

**Affiliations:** 1Department of Otorhinolaryngology and Head & Neck Surgery, University Medical Center Utrecht, Utrecht University, Heidelberglaan 100, 3584 CX Utrecht, The Netherlands; n.f.smithcortinez@umcutrecht.nl (N.S.-C.);; 2UMC Utrecht Brain Center, Utrecht University, Universiteitsweg 100, 3584 CG Utrecht, The Netherlands

**Keywords:** inner ear regeneration, endogenous progenitor cells, re-innervation

## Abstract

Sensorineural hearing loss is caused by damage to sensory hair cells and/or spiral ganglion neurons. In non-mammalian species, hair cell regeneration after damage is observed, even in adulthood. Although the neonatal mammalian cochlea carries regenerative potential, the adult cochlea cannot regenerate lost hair cells. The survival of supporting cells with regenerative potential after cochlear trauma in adults is promising for promoting hair cell regeneration through therapeutic approaches. Targeting these cells by manipulating key signaling pathways that control mammalian cochlear development and non-mammalian hair cell regeneration could lead to regeneration of hair cells in the mammalian cochlea. This review discusses the pathways involved in the development of the cochlea and the impact that trauma has on the regenerative capacity of the endogenous progenitor cells. Furthermore, it discusses the effects of manipulating key signaling pathways targeting supporting cells with progenitor potential to promote hair cell regeneration and translates these findings to the human situation. To improve hearing recovery after hearing loss in adults, we propose a combined approach targeting (1) the endogenous progenitor cells by manipulating signaling pathways (Wnt, Notch, Shh, FGF and BMP/TGFβ signaling pathways), (2) by manipulating epigenetic control, and (3) by applying neurotrophic treatments to promote reinnervation.

## 1. Introduction

Hearing loss is the most frequent sensory deficit in humans and is mainly caused by irreversible damage to cochlear sensory cells (hair cells) and/or their associated neurons (spiral ganglion neurons, SGNs). Irreversible hair cell loss is particularly caused by aging, noise exposure and ototoxic medication. In 2019, there were approximately 460 million individuals with disabling hearing loss, and according to the WHO, this number might increase to more than 900 million individuals by 2050 [1]. Hearing loss, which is often accompanied by tinnitus, results in high levels of morbidity, depression and social isolation, and it has been shown to significantly contribute to cognitive decline in the elderly [2,3,4,5,6,7,8]. While hearing aids and cochlear implants restore hearing in hearing-impaired and deaf individuals to a large extent, sounds are still perceived as distorted (for the sound of a cochlear implant, see [9] because the original cochlear function—in which the hair cells together with the basilar membrane play a key role—is not replaced. 

To prevent hair cell loss, there are currently several otoprotectants that can potentially prevent damage after ototoxic or noise-induced trauma, including antioxidants to reduce oxidative stress (such as sodium thiosulfate, amifostine, N-acetylcysteine) or anti-inflammatory medication, such as dexamethasone (for review on ototoxicity see [10] and for review on noise-induced hearing loss (NIHL) see [11]). Regeneration of lost cochlear cells has potential as an alternative approach and can have significant clinical applications to restore hearing without the need for an electronic device. Studies on the regeneration of the avian cochlea or zebrafish lateral line have demonstrated spontaneous hair cell regeneration out of endogenous progenitor cells after damage, even in adult specimens [12,13]. In contrast, the mammalian adult inner ear possesses limited regenerative potential [14,15]. The vestibular organ shows scarce but spontaneous hair cell regeneration after damage, whereas the cochlea shows no spontaneous regeneration [16]. For non-genetic hearing loss, targeting the endogenous cochlear progenitor cells by manipulating signaling pathways to promote hair cell renewal might improve the regenerative capacity.

Previous studies on the mammalian cochlea have mainly evaluated the presence of endogenous progenitor cells in the neonatal cochlea or in a normal hearing condition [17,18,19,20,21,22,23]. The effects of trauma, such as ototoxicity or noise exposure, on regenerative capacity of the cochlea of adult mammals has been scarcely studied. Because the majority of patients with SNHL are adults, studying the regenerative capacity after hair cell ablation in the adult mammalian cochlea is the key to understanding its true therapeutic potential. Importantly, the pathways that regulate cochlear development in mammals and hair cell regeneration in non-mammalian vertebrates shed light on the steps needed to improve mammalian hair cell regeneration in the future. In this review, we therefore summarize the development of the cochlea in mammals, necessary to understand the pathways involved in the generation of sensory hair cells and progenitors of the cochlea, as well as the presence of progenitors in the neonatal cochlea, and finally, we discuss the implications for the deafened adult mammalian cochlea. While previous reviews [24,25] mainly focused on hair cell regeneration, the current review not only evaluates the current literature on potential approaches to regenerate lost hair cells in the adult mammalian cochlea but also discusses re-innervation as that is crucial for functional improvement. For the purpose of this review, we focus on non-genetic hearing loss, considering that other treatment approaches are more likely to be effective in genetic hearing loss, in particular, gene therapy. 

## 2. Hearing Loss and Its Pathophysiology

### 2.1. Age-Related Hearing Loss

Age-related hearing loss (presbyacusis) is the most common sensory impairment in the elderly, and with aging of the population, the number of affected people is expected to rise rapidly [26]. Several age-related structural changes have been described, including age-related hair cell loss, SGN loss and atrophy of the stria vascularis [27]. Over the last decade, studies have also shown loss of inner hair cell (IHC) synapses and their afferent fibers [28,29]. 

Based on histopathology and patterns of hearing loss, Schuknecht et al. classified four main types of presbyacusis, including (1) sensory (hair cell loss at basal end of cochlea), (2) strial or metabolic (correlated to atrophy in the stria vascularis), (3) neural (as a result of loss of cochlear neurons), and (4) cochlear conductive or mechanical presbyacusis (due to stiffness of the basilar membrane) [29,30]. They concluded that the main contributing factor to presbyacusis was atrophy of the stria vascularis. Interestingly, more recently, it has been shown that presbyacusis is predominantly associated with damage to sensory cells, rather than age-related changes in stria vascularis [31].

The precise mechanism underlying age-related degeneration of different cochlear structures is unknown. Several contributing factors have been described, including inflammatory changes [32], genetic factors [33] and oxidative stress [34]. Although age-related changes are multifactorial, noise exposure is thought to be the major contributing factor of presbyacusis [31,35].

### 2.2. Noise-Induced Hearing Loss

NIHL mainly causes damage to and loss of outer hair cells (OHCs). Depending on the duration and intensity of the noise exposure, there may be IHC loss as well. Three mechanisms of noise-induced cochlear damage can be distinguished: (1) mechanical destruction by short exposure to extreme noise intensities causing direct trauma; (2) metabolic decompensation after noise exposure, which occurs over a longer period of time in high intensities; and/or (3) IHC synaptopathy leading to loss of SGNs [11,36,37]. Excessive noise stimulation causes the formation of free radicals or reactive oxygen species (ROS), as well as glutamate excitotoxicity, followed by activation of signaling pathways leading to cell death [38]. For an extensive review on cellular mechanisms involved in NIHL, see [36].

Apart from the loss of sensory cells, it has been widely investigated that noise exposure causes permanent damage to the ribbon synapses of the IHCs, also referred to as cochlear synaptopathy [37,39,40]. This leads to supra-threshold hearing loss, i.e., no measurable increase in hearing threshold but worse hearing at supra-threshold levels (including reduced speech perception in noise, hyperacusis) and tinnitus, also known as hidden hearing loss [37,40,41,42].

### 2.3. Ototoxicity

Ototoxicity is a pharmacological adverse reaction that causes irreversible damage to the hair cells in cochlear and vestibular tissue, leading to their functional loss. With over a million cases of profound ototoxicity-induced hearing loss annually worldwide, this is a major problem [43]. There are more than 600 categories of drugs registered with ototoxic side effects, and this number is still increasing [44]. The two most important ototoxic drugs are aminoglycosides (including gentamicin and kanamycin) and platinum-based antineoplastic agents (such as cisplatin, oxaliplatin and carboplatin). Numerous studies have been performed evaluating the effects of ototoxicity on the cochlea (for review, see [45]). Ototoxicity causes mainly OHC loss in a basal to apical gradient, thus associated with especially high frequency hearing loss. With higher concentrations or persistent exposure, ototoxic damage progresses to IHC loss as well. The mechanism by which ototoxic drugs affect the cochlea has not yet been fully elucidated. It has been suggested that oxidative stress induces apoptosis and necrosis in hair cells and marginal cells in the stria vascularis (for a review, see [46]). Ototoxins enter cells via active transport [47,48,49,50]. It has been recently shown that inflammation precedes oxidative stress and excessive production of ROS; therefore, it has been suggested that the inflammatory response triggers cell death [51]. After apoptosis of hair cells, the cell is extruded from the sensory epithelium and supporting cells phagocytize the remaining cell fragments. Supporting cells form a scar and preserve the epithelial cytoarchitecture and the integrity of the barrier of the organ of Corti [52,53,54]. Hair cell loss may occur rapidly (within days) after ototoxic exposure; following IHC loss, SGNs first become smaller and subsequently progressive SGN loss occurs [55]. The loss of SGNs has been associated with discontinued neurotrophic support from the organ of Corti [56,57].

Hair cell loss may occur rapidly (within days) after ototoxic exposure; following IHC loss, SGNs first become smaller and subsequently progressive SGN loss occurs [55]. Previous studies on ototoxicity have mainly focused on ototoxicity-induced hair cell loss and considered neuronal loss to be a secondary consequence caused by loss of trophic support [58,59,60,61,62,63]. However, as in NIHL, direct ototoxicity-induced damage to the synapse and SGNs may occur, as well as ototoxicity-induced swelling of the nerve [64,65,66,67].

## 3. Cochlear Development and Its Associated Pathways

For treatment approaches to regenerate lost hair cells after trauma, knowledge of inner ear development and pathways involved in hair cell and SGN formation is essential. For this section, we will focus on the development of both the human (in gestational weeks, GW) and the mouse cochlea (in embryonic days, E). 

Around GW6 [68] or E8-9 [69], the otocyst starts to develop from the otic placode. In the subsequent period (GW6-10 or E10.5-12.5), the cochlear duct develops, starting as an evagination of the otocyst [68,69,70,71]. In this process, several cells detach and form neuroblasts and SGN (and vestibular ganglion neurons, not further discussed in this review) precursors. Initial afferent innervation in the basal region of the cochlea is visible from GW8 [72] (E9.5), and proliferation and differentiation of SGN occurs in the basal-to-apical direction [72,73]. At GW12 (E13.5), SGNs extend their peripheral projections to the entire nascent organ of Corti [72,73]. This occurs when the first mature hair cells are visible in the basal turn (E13.5; refs. [70,71]) and when the scala vestibuli and scala tympani open and fill with fluid. The spiral form of the cochlea is finalized between GW10 and 11, reaching 2.5 turns [68]. Hair cells first start to differentiate from supporting cells around GW10 or E13.5 and then start to express their associated markers, such as Atoh1 [68,71]. After this, in the following days (in mice) or weeks (in humans), the cochlea matures from base to apex, following a spatiotemporal gradient of growth and inhibitory factors. Thus, the development of the basal turn is always ahead of the apical turn, and hair cell precursors that reside apically remain undifferentiated for a longer period than basally located precursors [71]. At GW14 (~E14.5), the last cells complete the cell cycle exit, and the characteristic organization of hair cells is visible: one row of IHCs and three rows of OHCs [68,70,71]. SGN neurons change morphology from branched to unbranched endings upon HC differentiation and maturation, forming radial bundles. At E18.5, their fibers extend throughout the whole basal OHC layer and by birth throughout the whole cochlea [73]. The cochlea further matures and differentiates up until GW19-20, when auditory function in humans starts [68]. For an extensive review on the development of mammalian cochlear development, see [74], and for a review on the development of afferent innervation, see [73,75,76].

### 3.1. Hair Cell Differentiation

Several important signaling pathways control hair cell development, of which the most important are Wnt, Notch, Sonic hedgehog (Shh), fibroblast growth factor (FGF) and bone morphogenetic proteins (BMP)/transforming growth factor β (TGFβ). These are further discussed below.

#### 3.1.1. Wnt and Notch Signaling

Wnt and Notch signaling in cochlear development have been extensively reviewed before [77,78]. Here, we discuss these pathways, as they are important for normal cochlear development and thus also for the regeneration of hair cells [79,80]. The canonical Wnt/β-catenin pathway plays an important role in fate determination, proliferation and differentiation of hair cells. The non-canonical planar cell polarity pathway (WNT-PCP) is important for cochlear lengthening and hair cell polarization, and it is the driver of stereociliary bundle formation and organization [77]. Activation of the canonical Wnt pathway leads to Sox2 expression and promotes cell proliferation [81]. In the early stages of cochlear development, Sox2 is present in all precursor cells [22,78] but is gradually downregulated in cells that do not retain supporting cell fate [82]. Through Wnt pathway activation, unphosphorylated β-catenin binds to TCF/LEF in the nucleus, causing transcription of Wnt target genes and expression of *Atoh1* [83]. Consequently, Notch signaling is downregulated, and these cells differentiate into hair cells [78]. In a process called lateral inhibition, increased Atoh1 expression in hair cell progenitors causes the expression of Notch ligands, which bind to Notch receptors in neighboring cells. Activation of *Hes1* and *Hes5* in these cells antagonizes *Atoh1* expression, leading to determination of their supporting cell fate [78]. These supporting cells activate Notch, which causes surrounding cells to attain their supporting cell fate, also known as lateral induction [78]. 

#### 3.1.2. Sonic Hedgehog (Shh) Signaling

Shh signaling has mainly been investigated in mouse and zebrafish studies [71,79,84,85,86]. Shh expression prevents hair cell differentiation during development. Shh signaling is initiated in the SGNs at E11.75. Through a decreasing Shh signaling gradient (from E13-15.5), starting at the base of the cochlea, hair cells start to differentiate, leaving apical prosensory cells undifferentiated at first [71,84]. Shh also inhibits retinoblastoma protein (Rb1), which, when active, is involved in cell cycle exit and, thus, final differentiation of hair cells [87]. It has been shown that mice that lack Shh have severe disturbances in cochlear development, such as the absence of the cochlear duct and undeveloped semicircular canals and endolymphatic duct [71,85]. Loss of Shh from developing SGNs during cochlear development causes premature hair cell differentiation, which impairs cochlear lengthening [86,88]. 

#### 3.1.3. Fibroblast Growth Factor (FGF) Signaling

The signaling of FGF is also important for cochlear development. It induces the development of the otic placode and the otic vesicle [89,90,91]. FGF receptors are expressed in the ventral part of the otic vesicle, where the cochlea will develop, both in the developing sensory epithelium and in adjacent mesenchymal cells [92,93]. Different FGF members regulate the development of non-sensory epithelium, hair cells, and pillar cell differentiation. FGF receptor expression is strong at the beginning of cochlear development and declines as the cochlea matures, where gradual attenuation of FGF is important for cell cycle exit and hair cell differentiation [89,92]. Depending on the localization of cells that express the FGF receptor, cochlear lengthening is also influenced by FGF, as has been shown in studies with FGF receptor deletions [89,93]. For a detailed description of FGF signaling in the mouse cochlea, see [94].

#### 3.1.4. Bone Morphogenetic Protein (BMP) and Tissue Growth Factor β (TGFβ) Signaling

BMPs belong to the TGFβ superfamily. In humans, mutations in TGFβ or its receptor are associated with hearing loss, such as in Loeys-Dietz syndrome and Ehler-Danlos syndrome [95,96]. In mice, BMPs are already expressed in early cochlear development from E9 onwards, together—but not always co-expressed—with the lunatic fringe (Lfng) [97,98,99,100]. The BMP pathway is necessary for prosensory domain specification and outer sulcus differentiation through phosphorylated SMAD expression [100,101]. Conditional knock-out of BMP receptors in the inner ear causes lack of cochlear development and lack of differentiation of cells in the prosensory domain, hallmarked by the absence of P27kip expression [100]. In developing chicks, BMP signaling inhibition by Noggin causes an increased size of sensory patches without changes in cell proliferation in E3.5-4 [99] but a reduction in hair cells and supporting cell counts in E15-16 [96]. On the other hand, the addition of exogenous BMP4 in developing chick otic vesicles in vitro showed a reduction in sensory patches and a marked decline in cell proliferation in E3.5-4 [99] but an increase in hair cell counts in E15-16 [97]. High concentrations of a BMP agonist downregulate Lfng and FGF-10 expression in cochlear cultures [100]. Considering this for optimal hair cell regeneration, combining BMP activation in the early stage of development and BMP inhibition in the later stage might be necessary, corresponding with the previously published protocol in inner ear organoid generation from induced pluripotent stem cells [102]. 

#### 3.1.5. The Role of Micro-RNAs in Cochlear Development

In addition to signaling pathways, micro-RNAs (miRs) play an important role in cochlear development. miRs are non-coding (very small strings) of RNA that play important roles in post-transcriptional regulation of gene expression. miR are first expressed in sensory neurons from day E12.5 and in hair cells from day E14.5 in hair cells [103]. They depend on *Neurog1* and *Atoh1* expression, respectively. miR follows a basal-apical gradient during maturation, which also marks cochlear development in several other signaling pathways [103,104]. Knock-out of miR during several stages of cochlear development showed impairments in hair cell development, such as hair cell death after defective hair cell bundle morphology [104]. Well studied miR are miR-183, expressed during sensory neuron and hair cell development, miR-96 of which a mutation causes progressive hearing loss in humans [103,104], and miR-124, which controls the Wnt pathway through manipulation of secreted frizzled-related protein 4 (Sfrp4) and Sfrp5 [105]. In this review, we will not further discuss miR in hair cell regeneration because our focus is on signaling pathway regulation.

In summary, hair cell differentiation is dependent on many factors and pathways that are up- or downregulated during cochlear maturation in a base-to-apex gradient. We describe the most important drivers: Wnt, Notch, Shh, FGF, and BMP/TGFβ, which through their role in development, are important in research on hair cell regeneration in the cochlea after damage.

### 3.2. Spiral Ganglion Neuron Differentiation

Separation of the anterior and posterior region of the otocyst is directed by Retinoic acid (RA), and opposite gradients of Wnt and Shh assist the assignment of the proneurosensory region in the anterior (ventral) part [73]. SGN formation and differentiation occur in the anterior region. Lfng, Six1, Eya1 and Sox2 are important markers in this region [73]. Both Neurogenin (Ngn-1) and Neurod1 expression are important for neurogenic maturation, which occurs through negative feedback on Sox2 by these markers. SGN development is driven by neurotrophins secreted by the otic epithelium, cochlear Schwann cells and supporting cells (and in later stages, SGN cells). The most important neurotrophins are BDNF and neurotrophin-3 (NT-3), which are linked with transcription factors neuroD1, Eya and Brn3a. In the developing cochlea, Bdnf expression migrates from supporting cells in the apex to hair cells in the base, and NT-3 expression is expressed in supporting cells but later expands to IHCs. Neurotrophins activate TrkB and TrkC receptors to mediate SGN survival, and they are expressed in opposite gradients [73,106]. The absence of NT-3 expression causes massive neuronal loss in the base, while the absence of Bdnf leads to a relatively small reduction in the apex (reviewed in [56,106]). The final location of SGNs and their correct locational outgrowth are highly dependent on several pathways. Semaphorins, POU3f4 (through ephrins), plexins/neuropilin, slits and netrin (receptors) are important for axon guidance and target innervation by limiting axonal outgrowth, ensuring that they do not overshoot beyond OHC layers. Directional growth is mediated by Prox1 and R-spondin2. Ephrins (B2 and A4) regulate neurite outgrowth in the cochlear epithelium through ERK signaling and are involved in the formation of radial bundles of SGNs. Finally, pillar cells—driven by FGFR3—divide the IHC from OHC, guarding which SGNs can cross to the OHC layer and which cannot [73]. SGN formation and differentiation occur near parallel to hair cell development and differentiation. As HC differentiation starts later than SGN development, it is influenced by this process, among others, by Shh [71]. At first, through random action potentials, hair cells form synapses with several SGNs, depending on the location within the cochlea. SGN connected to HC form radial bundles that project to the central nervous system. Postnatally, 50% of the developed synapses are reduced through synaptic pruning. The remaining synapses further mature, now called ribbon synapses, mediated by thyroid hormones, enabling calcium signaling and neurotransmitter release. While during development, two types of SGN (type-I and type-II) are connected to both IHC and OHC, perinatally, SGN-type-II synapses disconnect from IHC and conversely SGN-type-I from OHC, enabling SGN-type-II to connect to multiple OHCs. This leaves IHC connected to type-I (~95%) and OHC to type-II SGN (reviewed in [73]). 

In summary, final SGN formation and location are reached through four important steps. (1) SGN development is initiated by neurotrophins BDNF and NT-3, secreted by cochlear Schwann cells and supporting cells. (2) Neurite outgrowth and axon guidance to sensory epithelium, driven by ephrins through POU3f4, semaphorins (Sema3a-g), neuropilin/plexins, slits (robo receptors) and netrin (receptors). (3) SGN innervation of HCs, including ribbon synapse formation, driven by thyroid hormones (calcium signaling). (4) Synaptic pruning, mostly postnatally, and disconnection/retraction of redundant SGN connections to IHCs and OHCs. For a detailed description of the signaling pathways and transcription factors controlling SGN differentiation, see [76]. 

## 4. Target Cells for Potential Regenerative Treatment

Of special interest in cochlear development and hair cell regeneration are several modulators of Wnt signaling: members of the Lgr family (mainly Lgr4, -5 and -6). Lgrs are stem cell markers expressed in many types of adult stem cells and have gained attention through their potential in regenerative therapies [107,108]. In the cochlea, cells expressing Lgr5 or Lgr6 have the capacity to divide and differentiate into hair cells in vitro or in vivo [20,22,109]. Of all Lgrs, Lgr5 expression is the most well described, as a mouse strain harbors an Lgr5-EGFP-IRES-creERT2 knock-in allele that allows for easy visualization of Lgr5-positive cells with a GFP reporter system [107]. This mouse model is widely used in studies on the regenerative capacity of the cochlea [23,82,109,110,111]. At E17, Lgr5 is expressed in the whole organ of Corti, in all hair cells and supporting cells from base to apex. During further development, Lgr5 expression gradually decreases, and in the first week of birth (P1), Lgr5 is expressed in the greater epithelial ridge (GER), the third row of Deiters’ cells, inner border cells and inner pillar cells [22,23,82]. From then, expression further decreases, and at P30, Lgr5 is restricted to the third row of Deiters’ cells and, to a lesser extent, the inner pillar cells and expression of Lgr5 is then stable up until at least P100 [22,23,82,110]. In early development of the mouse cochlea, Lgr4 is expressed in the pro-sensory domain, non-sensory domain and SGNs of the developing cochlea. At E17, Lgr4 expression is particularly strong in differentiating hair cells, where it is co-expressed in Lgr5-positive cells [23]. In the mature cochlea (from P21), Lgr4 is restricted to all three rows of Deiters’ cells [23]. Lgr6 is expressed in the inner border cells and inner pillar cells in the developing cochlea until P15, after which it is only expressed in inner border cells at P20 and decreases until it is no longer expressed at P30 [20]. 

Another marker used to label progenitor cells is Lfng, a type of Fringe gene involved in cochlear development. It is already expressed in an early stage of development in a base-to-apical gradient in the developing prosensory domain in the region where later hair cell differentiation takes place [69]. Lfng is first expressed in both supporting cells (at Kölliker’s organ and the lateral border of the developing organ of Corti) and in the area where hair cells differentiate [100,112]. In the adult cochlea (P60), Lfng is restricted to supporting cells only, including the inner phalangeal cells, pillar cells, and all three rows of Deiters’ cells [100,112,113]. 

The epithelial cellular adhesion molecule EpCAM (or CD326) is a protein that participates in cell–cell adhesion, proliferation and maintenance of undifferentiated tissues, as well as regulation of differentiation, migration and invasion [114]. In the cochlear duct, it is expressed in all sensory and non-sensory epithelial cells during development [115]. It is used in vitro as a marker to isolate hair cell progenitors from the prosensory domain of neonatal and adult mice and the fetal human cochlea [115,116]. 

Sox2 is a transcription factor that is involved in neurogenesis and in the proliferation and maintenance of stem cells, including neural stem cells [74]. Sox2 is required for the development of the sensory domain of the inner ear, and it is expressed in embryonic stages in hair cells, adjacent supporting cells and the GER [117,118]. Sox2 marks sensory progenitors early in inner ear development and acts upstream of the Atoh1 gene during sensory organ formation [119,120]. Postnatally, it is expressed in all supporting cells [19,22] and in the mature cochlea is present in border cells, inner phalangeal cells, inner and outer pillar cells, Deiters’ cells and Hensen’s cells [21,121]. 

In summary, Lgr5, Lfng, EpCAM and Sox2 are expressed in the adult cochlea, and they represent potential target cells for future regenerative therapies. An overview of the organization of the adult cochlea and the markers of supporting cells with stem cell properties is listed in Table 1 and shown schematically in Figure 1.

## 5. Regenerative Potential of the Cochlea

### 5.1. Non-Mammalian Vertebrates

Hair cell regeneration is readily observed in non-mammalian vertebrates, especially birds and fish. Understanding the differential response to damage is important for making the translation to mammalian regeneration. In the avian cochlea, regeneration of hair cells leading to recovery of function is observed 1 or 2 months after ototoxic or noise-induced damage [13]. Avian cochlear supporting cells have progenitor cell-like characteristics and express progenitor markers, including Sox2, which directly activate Atoh1 to induce hair cell fate [123]. After ototoxicity, new hair cells are generated from Sox2+ supporting cells, with preservation of the total number of supporting cells [124]. Supporting cells either mitotically divide or directly transdifferentiate to produce new hair cells [12]. Direct transdifferentiation from supporting cells to hair cells has been observed as early as 15 h after damage with gentamicin, reaching complete maturation 2 days after damage. In contrast, mitotic proliferation of supporting cells occurs 2 to 3 days after damage, and they continue to proliferate for 3 weeks [12]. In birds, two distinct groups of supporting cells give rise to tall (similar to IHCs) or short (similar to OHCs) hair cells and have distinct gene expression patterns [125]. Comparative analysis between supporting cells in birds and mammals should further clarify to which mammalian supporting cells these two distinct supporting cell populations in bird cochlea relate.

Transcriptomic analysis in ototoxically damaged cochlea has allowed us to find up- or downregulated genes that might contribute to the signaling during regeneration. In neonatal chickens treated with gentamicin to induce complete hair cell loss, transcriptomic analysis revealed that the Wnt, Notch, FGF and BMP/TGFβ signaling pathways were differentially expressed [126,127]. This confirms that these pathways control avian inner ear regeneration, either by up- or downregulating genes [126,127]. In line with these findings, it has been shown that manipulation of Notch, Wnt, BMP/TGFβ or FGF signaling pathways strongly upregulate hair cell regeneration after hair cell death in avian cochlear explants [126,127,128]. Moreover, Bai et al. showed that simultaneously inhibiting the Notch (with DAPT) and BMP (with Noggin) signaling pathways and activating the Wnt (with LiCl) signaling pathway has the strongest effect on promoting hair cell regeneration, suggesting that combined strategies are most effective [128]. Further analysis of regenerating basilar papilla in chicks revealed that supporting cells upregulate immune-related genes via the JAK/STAT signaling pathway [127,129]. These studies confirmed that the JAK/SAT signaling pathway is also necessary for hair cell regeneration [127,129]. 

The zebrafish is another experimental model for hair cell regeneration since it expresses sensory hair cells not only in the inner ear but also on the skin in the lateral line organ. This superficial presence of sensory hair cells allows for rapid drug-induced damage and live imaging to assess cell death and/or regeneration in vivo [130,131]. Interestingly, 15–20 min after ototoxic trauma, there is complete hair cell ablation in the skin of zebrafish with complete regeneration of hair cells within 72 h after damage. In zebrafish, after ototoxicity, surviving supporting cells produce new hair cells mainly by mitotic regeneration, and regeneration-specific genes are upregulated as early as 30 min after ototoxicity [132,133,134,135]. These studies have revealed a distinctive population of cells in the zebrafish lateral line, other than supporting cells, that act as quiescent stem cell reservoirs for non-homeostatic regeneration [135,136]. 

As in avian inner ear regeneration, the Wnt, Notch, FGF and BMP signaling pathways also control supporting cell proliferation and differentiation into hair cells in zebrafish [132,136,137,138]. Notch, FGF and retinoic acid (RA) signaling are downregulated 1 h after ototoxicity-induced hair cell damage; BMP signaling is upregulated 1 h after damage and Wnt signaling is activated only 5–12 h after damage in zebrafish [132,136,137]. Indeed, it has been demonstrated that the trigger to induce Wnt signaling is the downregulation of Notch signaling after hair cell loss in zebrafish [132,137]. Interestingly, it has been concluded that the absence of immediate and transient Notch downregulation in the mammalian cochlea is a crucial difference with the zebrafish lateral line and therefore temporary inhibition of Notch is potentially an important first step toward mammalian hair cell regeneration [137].

Although the importance of Shh signaling in cochlear development has been studied in mammals and non-mammals, little is known about the role of Shh in hair cell regeneration after trauma in non-mammalian vertebrates. As in the mammalian cochlea, it potentially plays a role in development and tonotopic organization [85,139]. 

### 5.2. Neonatal Mammalian Cochlea

The neonatal mouse cochlea possesses the capacity to produce new hair cells from supporting cells, such as inner pillar cells and Deiters’ cells, by targeted Atoh1 ectopic expression [140,141]. For a review on reprogramming transcription factors (e.g., Atoh1, Gata3, POUF1) as a tool to regenerate sensory hair cells, see [142]. A subtype of supporting cells that express Lgr5 has progenitor potential and can regenerate into new hair cells in vitro and in vivo [17,18,19,80,122,143,144,145,146,147]. After trauma to the cochlea, neonatal supporting cells have been shown to survive and even retain regenerative potential in vitro, ex vivo (cochlear explants) and in vivo [17,109,146,148]. Further, it has recently been described that spontaneous regeneration of hair cells in damaged neonatal cochlea produces cells with markers for innervated IHCs and OHCs [149]. Lgr5-expressing supporting cells mainly generate outer hair cells after both IHC and OHC loss [17]. Moreover, it has been shown that Lgr5+ cells show a greater capacity for OHC regeneration after ototoxic trauma compared to Lgr5+ cells that were not stressed with ototoxic medication [145]. Interestingly, transcriptomic analysis of neonatal mouse cochleae after gentamicin-induced trauma revealed that, similar to the avian cochlea, genes in the Wnt, Notch, FGF, BMP/TGFβ and Shh signaling pathways were differentially expressed [150]. Moreover, neomycin-treated cochlear explants showed upregulation of Fgf3 and Egfr (confirming these signaling pathways control inner ear regeneration in mammals) and downregulation of Hes1 and Hes5 (suggesting Notch signaling inhibition) besides a clear upregulation of genes involved in the cell cycle, suggesting neomycin induced the proliferation of Lgr5+ cells [145]. It has been demonstrated that inhibition of TGFβ signaling via overexpression of follistatin alone or in combination with LIN28B allows supporting cells to re-enter the cell cycle and produce mature hair cells after P5 in mice [151], suggesting inhibiting TGFβ signaling might be crucial to adult hair cell regeneration. In recent studies, a new progenitor cell population has been described in the great epithelial ridge (GER) of the neonatal mouse cochlea; after selective ablation of Lgr5+ SCs, cells in the GER mitotically divide and migrate to repopulate the organ of Corti with mainly inner phalangeal cells [152]. 

### 5.3. Adult Mammalian Cochlea

Only a few studies have evaluated the regenerative capacity of the adult cochlea. It is thought that the regenerative potential of the cochlea of adult mammals is less than that of neonatal mammals. It has even been suggested that the regenerative capacity of supporting cells is lost in the adult cochlea [18,21,153]. However, White et al. showed that although the proliferative capacity of supporting cells is decreased with age, supporting cells from the mature adult cochlea can generate hair cells in vitro [153]. It is known that supporting cells are more resistant to an ototoxic event compared to hair cells, which is probably a result of less uptake of aminoglycosides in supporting cells [154,155,156,157]. In line with this, it has been shown that Sox2+ supporting cells survive 6–12 months after ototoxic trauma in mice, even after complete ablation of IHCs and OHCs [14,21,121]. This is in line with what is shown in NIHL, where Sox2+ supporting cells are also more resistant to noise trauma [148]. Several other studies support the hypothesis that supporting cells with regenerative capacities are present in the adult cochlea, which survives after trauma [158,159]. Moreover, we have shown that Lgr5+ supporting cells survive severe ototoxic trauma in adult Lgr5GFP transgenic mice [110]. Interestingly, cell proliferation has also been reported after high concentrations of ototoxic medication in the mature cochlea [159]. Although these studies suggest the availability of endogenous cochlear stem cells as targets for treatments, even after trauma, it was not determined if the surviving supporting cells had any progenitor capacity (e.g., by evaluating if surviving supporting cells could produce hair cells in vitro). More studies are needed to investigate whether and to what extent surviving supporting cells, including Lgr5+ cells, maintain regenerative potential. For the assessment of their potential for clinical applications, future studies are also necessary to unravel the long-term effects of trauma on regenerative potential. 

### 5.4. Epigenetic Barrier to Hair Cell Regeneration in Adult Cochlea

Epigenetic modifications during the development of the cochlea promote gene transcription associated with regeneration in neonatal stages (for an overview, see [160,161]); however, repressive complexes limit cochlear regeneration in adulthood [162,163,164]. Histone modifications and DNA methylations are the main epigenetic modifications controlling the access of transcription factors to chromatin. In Lgr5+ inner ear stem cells, modifications in histone H3K4me regulate proliferation and hair cell regeneration after neomycin treatment [165]. Interestingly, inhibition of histone deacetylases (HDAC) with valproic acid (VPA) has been shown to promote hair cell differentiation in vitro [109]. In addition, inhibition of DNA methylation (with 5-azacytidine) has been shown to promote hair cell differentiation in Sox2+ supporting cells in adult mice [166]. Lastly, inhibition of demethylation in histone H3K4me (with GSK-LSD1) significantly increased the number of supporting cells that transdifferentiated to hair cells [162]. Thus, breaking the epigenetic barriers of the endogenous cochlear stem cells might be the key to improving hair cell regeneration in the adult cochlea.

## 6. Promoting Regeneration & Re-Innervation

As reviewed here, important signaling pathways (Wnt, Notch, Shh, FGF, BMP/TGFβ) in the zebrafish lateral line and in the avian cochlea have been described to participate in hair cell regeneration, and these same pathways are known to control otic development in the mammalian cochlea. For that reason, in the last decade, novel strategies have focused on targeting these signaling pathways in the mammalian cochlea to improve hair cell regeneration.

### 6.1. Manipulating a Single Signaling Pathway

Several studies using neonatal mammalian cochlear explants have shown that inhibition of Notch signaling by pharmacologic agents or gene therapy results in hair cell regeneration from supporting cells [17,150,167,168,169,170,171]. Bai et al. showed that in undamaged neonatal cochlear explants, inhibition of Notch generated increased OHC numbers. Treatment of gentamicin-damaged neonatal cochlear explants with a Notch inhibitor (DAPT) also resulted in newly formed (but disorganized) hair cells with no increased proliferation rates, suggesting supporting cells directly transdifferentiated to hair cells to repopulate the cochlea [168]. In addition, inhibition of Notch by a novel gamma-secretase inhibitor (CPD3) showed increased OHCs by direct transdifferentiation of supporting cells, in bacterial-induced ototoxicity in neonatal whole otic capsule explants [169]. In contrast, in neonatal neomycin-damaged cochleae Notch inhibition (with Dibenzazepine or DAPT) also strongly promoted supporting cell proliferation, in addition to hair cell regeneration, and upregulated Wnt and Atoh mRNA expression [171,172]. Thus, CPD3 allowed hair cell formation by direct transdifferentiation of Sox2+ and Pouf4+ cells, whereas Dibenzazepine and DAPT induced hair cell formation by proliferation of Sox2+ supporting cells. It must be taken into account that CPD3 was tested in otic capsule explants derived from P2 Wistar rats, while Dibenzazepine and DAPT were tested in cochlear explants from P1 mice; these differences in models used could also explain the promotion of proliferation of the last two drugs versus CPD3; however, Dibenzazepine and DAPT were able to downregulate Notch and activate Wnt signaling, thus promoting proliferation, and CPD3 showed downregulation of Notch without Wnt activation, and no proliferation was observed. These results confirm the importance of activating the Wnt pathway in order to promote supporting cell proliferation prior to or alongside hair cell differentiation by Notch inhibition. 

In undamaged conditions, Wnt signaling activation promotes the proliferation of supporting cells and the generation of new hair cells [19,150,173]. However, the production of new hair cells after Wnt activation depends on the developmental stage in which they are treated, as new hair cells could be generated at E13.5, but not at E16 in mouse cochlear explants [173]. In neomycin-induced hair cell loss, Wnt activation alone is not able to induce supporting cell proliferation or hair cell differentiation [171].

Increased hair cell regeneration has also been observed after in vitro induction of the Shh signaling pathway in ototoxically damaged cochlear explants [87,147]. In P2 rat cochlear explants, Shh stimulation after neomycin-induced damage promoted the proliferation of Sox2+ supporting cells and the generation of new hair cells [87]. In addition, Lu et al. showed that Shh also triggers the proliferation of supporting cells and hair cell regeneration [87]. They suggested that this was a result of an increase in cell cycle-entry related genes mediated by the inhibition of retinoblastoma protein [87]. In line, Chen et al. showed that activation of the Shh signaling pathway promoted supporting cell proliferation and mitotic hair cell regeneration after neomycin-induced hair cell loss in P1 mice [147]. Interestingly, Chen et al. showed that Shh activation promoted both Wnt signaling activation and Notch signaling inhibition, hence supporting the hypothesis that Wnt activation is crucial for hair cell regeneration [147]. Still, more studies are needed to evaluate whether targeting Shh allows hearing recovery as well as de novo hair cell production.

As we have observed here, manipulation of a single signaling pathway is probably not sufficient to support cell proliferation and hair cell formation to allow clinically relevant functional outcomes. More importantly, only two studies have evaluated hair cell regeneration in adult specimens. In undamaged adult mouse cochleae, transient Notch inhibition enabled supporting cells to proliferate after treatment with Atoh1 gene therapy and efficiently transdifferentiate into hair cell-like cells in vitro [174]. In another report using adult mice deafened by acoustic trauma, Notch inhibition alone promoted supporting cell transdifferentiation to OHCs, but with slightly improved ABR thresholds of only 8 dB, these animals were still functionally deaf [148]. 

### 6.2. Combined Strategies

Novel combinational strategies targeting endogenous progenitor cells, such as Lgr5+ cells, have shown promising results in promoting hair cell regeneration in vitro. For example, a combination of Wnt activators and Notch inhibitors resulted in extensive supporting cell proliferation and hair cell formation in neonatal cochlear explants damaged by neomycin [142]. Samarajeewa et al. showed in E13.5 and P0 cochlear explants that Wnt activation and Notch inhibition promoted supporting cell proliferation and hair cell formation [170]. Interestingly, Wu et al. showed that adding a Shh agonist to the treatment with a Notch inhibitor (DAPT) and a Wnt agonist (QS11) after neomycin-induced damage in neonatal cochlear explants improved supporting cell proliferation and hair cell production, compared to manipulating only Notch and Wnt (or Notch alone) [171]. This supports not only the hypothesis of the additional value of combined strategies but also shows the importance of inhibiting the Shh pathway to improve hair cell regeneration. In addition, Wnt activation (CHIR99021) in combination with inhibition of TGFβ signaling by an antibody targeting the vascular endothelial growth factor receptors (regorafenib) showed potential to induce hair cell regeneration in damaged neonatal cochlear explants [175]. Interestingly, the combination of Wnt activation (CHIR99021) and FGF inhibition (Su5402) triggered an increase in IHCs but at the expense of the inner pillar cell pool [173]. Moreover, Wnt activation in combination with a histone deacetylase inhibitor, which promotes epigenetic manipulation, as well as Notch inhibition, resulted in extensive supporting cell proliferation and both IHC and OHC formation in neonatal cochlear explants [109]. 

These observations suggest that manipulating a combination of key signaling pathways to target endogenous cochlear progenitors, including epigenetic manipulation, is a potential therapeutic approach for hearing loss in adults. Unfortunately, this has mainly been investigated in the neonatal cochlea or ex vivo and the functional effects are still unknown. Although cochlear explants represent a plausible model for hair cell regeneration in vitro, they lack the ability to measure functional outcomes to reveal whether the newly formed hair cells are able to transmit sound in a physiological environment. For that, it is important that we study the effect of these drugs in vivo using animal models and analyze functional improvements, if any.

Whether and to what extent adult Lgr5+ supporting cells or other supporting cells with progenitor capacity retain responsiveness to Wnt activation, Notch inhibition and/or to the manipulation of other signaling pathways after deafening to improve supporting cell proliferation and hair cell regeneration remains unresolved and should be the focus of future research. To improve hearing, in particular in the case of severe hearing loss, promotion of IHCs, and not only OHCs, is crucial since IHCs are the ones to transfer the acoustical information to the auditory nerve, whereas OHCs ‘only’ act as amplifiers, increasing sensitivity and frequency tuning. Therefore, further research in the adult cochlea is needed to evaluate the efficacy of OHC and IHC regeneration.

### 6.3. Improving Functional Outcomes: Inner Hair Cell Re-Innervation after Cochlear Trauma

To improve the restoration of hearing in deafened patients, re-innervation of newly generated hair cells and re-formation of the lost synaptic connections, known as ribbon synapses, are necessary. By focusing the efforts on a multi-targeted approach, the chances of restoring hearing after cochlear damage are potentially higher, giving the patient a better opportunity for recovery. 

Protection of SGNs by neurotrophins after cochlear damage has been widely studied (for review, see [56]). The four types of neurotrophin nerve growth factor (NGF), BDNF, NT-3, and NT-4/5 participate during cochlear development, and exogenous treatment has been shown to ameliorate SGN loss and improve the number of ribbon synapses after cochlear damage in animal studies [176,177,178,179,180,181]. Other novel therapies involve drugs targeting different molecules in the cochlea, such as the repulsive guidance molecule (RGMa) with antibodies or using Trk receptor agonists; these strategies have been shown to promote neuronal survival, restore synapse after damage and even restore hearing to control levels in adult mice [182,183,184]. Another candidate to reverse cochlear synaptopathy is zoledronate, a bisphosphonate used in the clinic to treat osteoporosis and related bone diseases, which has also been shown to reverse the damage caused by noise exposure and promote the recovery of hearing in adult mice [185]. Finally, treatment with exogenous insulin-like growth factor (IGF) restored the number of ribbon synapses in the cochlear explants after exitotoxic trauma [186]. Interestingly, in patients with sudden deafness, IGF promoted hearing restoration (10 dB improvement; refs. [187,188]. Moreover, it has been recently shown that overexpression of neuritin (a gene induced by NT-3 and BDNF; ref. [189]) is able to mitigate the damage to hair cells and even promote supporting cells to transdifferentiate to hair cells after gentamycin-induced damage to cochlear explants derived from adult mice [190], so combining these treatments could prove successful not only to help re-innervation but also to promote hair cell regeneration. The additional value of adding neurotrophins (or other neuroprotective/neuropromotive drug) to the regenerative treatments targeting endogenous cochlear progenitor cells on functional outcomes needs to be further explored.

## 7. Human Inner Ear Regeneration and Clinical Trials Targeting Endogenous Stem Cells

In humans, there is little evidence of inner ear regeneration. Three-dimensional cultures have allowed the expansion and experimentation of human-derived cochlear organoids. Roccio et al. showed that fetal-derived post-mitotic EpCAM+ cells were able to proliferate, expand and generate hair cells in vitro. After differentiation with a Wnt activator (GSK3β inhibitor CHIR99021) and a γ-secretase (Notch) inhibitor (LY411575), there was no increase in the number of hair cells generated compared to expansion medium alone, suggesting there was some spontaneous transdifferentiation in hair-cell-like cells in the expansion medium conditions [115]. 

McLean et al. showed for the first time that human adult-derived inner ear organoids were readily generated using the same protocol as for growing mouse cochlear organoids, which include Wnt agonists and Notch inhibitors, an extracellular matrix (e.g., Matrigel, to produce a 3D environment to sustain their growth) and growth factors (e.g., insulin-like growth factor, epidermal growth factor, fibroblast growth factor) [109]. These organoids have differentiation potential and can generate hair cells in vitro. The tissue was collected from only one patient, so further experiments are needed to explore the conditions for growing human (adult)-derived inner ear organoids. 

In another study using inner ear tissue from adult patients, Senn et al. showed that adult-derived cochlear and utricular spheres grew in vitro [191]. However, they showed that only 1 in 10 postmortem cochleae could generate organoids, which was less successful compared to the vestibular epithelium-derived organoids (success rate > 50%). This is in line with the fact that the vestibular system has more regenerative capacity than the cochlea in adulthood [16]. The low yield of organoid generation achieved in the Senn et al. study could be explained by the absence of the key Wnt agonists and Notch inhibitors and the lack of a 3D environment in their culture conditions. 

Although research on human inner ear regeneration has just started, some researchers have initiated clinical trials targeting either cochlear endogenous progenitor cells or ribbon synapses as a treatment for SNHL in adults (for an extended review on this topic, see [192,193]). The REGAIN clinical trial (EudraCT number 2016-004544-10) reported positive results from a phase I multiple ascending dose, open-label safety study of a novel gamma-secretase (Notch) inhibitor (LY3056480) in 15 patients with mild to moderate SNHL. The REGAIN clinical trial has continued to treat patients with SNHL and will evaluate pure tone hearing thresholds and speech-in-noise perception. Furthermore, a pioneer clinical trial has already shown that modulating Wnt (CHIR99021) and Notch (LY411575) signaling pathways with a recently commercialized drug (FX322) in patients with SNHL improves speech recognition in-quiet and in-noise 90 days after treatment [194]. Several clinical trials are ongoing to test the efficacy of FX322 to improve hearing in stable, acquired, adult-onset SNHL associated with noise-induced, idiopathic sudden, and age-related SNHL [192]. No results have yet been published on these ongoing clinical trials. Still, further combined strategies, as discussed above, are probably necessary to improve functional outcomes. 

## 8. Conclusions and Future Perspectives

In this review, we have discussed the therapeutic potential of regenerating hair cells from remaining endogenous cochlear stem cells for acquired non-genetic SNHL in the adult cochlea. Current research mainly focuses on regeneration in the neonatal cochlea, where different types of supporting cells, including Sox2+ and Lgr5+ supporting cells, allow the opportunity for regeneration of hair cells. Whereas other reviews on the regenerative potential in the adult cochlea focus on specific pathways regulating regeneration or epigenetics [24,25], this review summarizes current knowledge and literature on regeneration in the adult cochlea through the different types of supporting cells known and their regenerative potential. Furthermore, we propose a combined strategy for regeneration, based on manipulation of key signaling pathways (e.g., Wnt, Notch, Shh, FGF, and BMP/TGFβ), manipulation of epigenetic modifications to expose transcriptional sites, and for functional effects, as well as treatments to protect and regenerate the SGNs and ribbon synapses (including neurotrophic factors, and/or TrkB receptor agonists, anti-RGMa and IGF).

The pathways controlling development of the cochlea were described and studies on hair cell development and regeneration in the cochlea were discussed. Knowledge of these pathways poses a better understanding of the possibilities within regeneration, such as the use of Wnt-, Shh-, FGF- agonists or Notch- and BMP- inhibitors. They target supporting cells with stem cell capacity, such as Lgr5- and Sox2-positive cells, and induce hair cell differentiation. These supporting cells are the main drivers of hair cell regeneration in the neonatal cochlea. However, the regenerative potential of the adult cochlea has yet to be investigated, since it is of high importance in regenerative medicine, as onset of hearing loss often occurs during adulthood. There is evidence that these endogenous cochlear stem cells survive to a certain extent after noise exposure or ototoxicity, even in the adult cochlea, which indicates the availability of target cells for future therapies. However, the regenerative capacity of these stem cells after deafening needs to be further evaluated. 

Several models can be used to test the efficacy and safety of the different drugs manipulating the key signaling pathways: in vivo approaches using zebrafish and different rodent models, such as the Lgr5-GFP mouse model [19,22,110], or ex vivo cochlear explants [146,173,186]. These are also important models for investigating the mechanisms of regeneration and the effects of possible modulators. Moreover, in vitro expansion and culturing of supporting cells has been an increasing field of interest, now that supporting cells can be easily isolated, for example, through sorting for surface markers, such as EpCAM, or through GFP-sorting, isolating Lgr5-positive cells from the Lgr5-GFP transgenic mouse [109,115]. From these isolated cells, 3D cultures can be generated into inner ear organoids, which can be used to test many different protocols manipulating the key signaling pathways in order to evaluate how to improve the regeneration of hair cells [22,109,115,169,175]. Recently, a study investigating different drugs and small molecules on these organoids using high-throughput screening led to insights into small molecules that cause hair cell differentiation [175]. For translational purposes, human cochlear organoids from the adult cochlea are important to use, but to grow them is still a challenge and further research is needed to evaluate whether that can be improved. Adding certain growth factors (e.g., IGF, epidermal growth factor, FGF, antioxidants, Wnt agonists) can help to improve the growth of organoids. 

After optimization of the differentiation protocol in organoids, functional outcomes can be tested in animal models. Animal studies also allow us to evaluate the additional value of neurotrophic factors, such as BDNF, NT-3 and/or TrkB receptor agonists, anti-RGMa, and IGF to improve SGN survival and the regeneration of the ribbon synapse. Another model to test functional connections between newly formed hair cells and SGNs could potentially be microfluidic-based approaches, which can establish spatially controlled cellular structures in the form of an organ-on-a-chip [195]. However, although this technology has been used for many organs such as liver, heart and kidney for drug toxicity assays and even multi-organ systems to test systemic or off-target effects, no inner ear-organ-on-a-chip has been developed yet.

The first clinical trials modulating the Wnt and/or Notch pathways with novel compounds are currently ongoing [109]. These trials use intratympanic injections; however, while this route is clinically feasible, the optimal route of delivery from a therapeutic perspective is still unknown. Intratympanic delivery has a risk of low yield of medical compounds that enter the cochlea. Other delivery methods, such as intracochlear or systemic (IV), have several disadvantages as well [196]. Intracochlear injections have a risk of infection and increased cochlear trauma, causing further damage. For systemic application, there is a risk of systemic side effects. Additionally, the blood-labyrinth barrier must be overcome, and as a result of premature systemic clearance, only low concentrations reach the cochlea. Minimizing invasiveness and maximizing the efficacy of treatment have to be carefully balanced. More experimental methods are currently being investigated, such as the application of gelatin sponge on the round window in the middle ear or the tympanic membrane [197,198,199], microperforations of round window membrane to increase the rate of diffusion into the cochlea [200] or magnetic targeting to improve the concentration of systemically delivered compounds in the cochlea [201]. 

Future research should be directed toward the assessment of a tailor-made approach for hair cell regeneration of the adult (damaged, mammalian) cochlea. Nevertheless, with the current understanding of signaling pathways and epigenetics involved in development and regeneration of the cochlea, we propose a combined approach, implementing multiple targets to induce regeneration of hair cells, re-innervation of newly generated hair cells and re-formation of the lost synaptic connections, which is probably the key to success. 

## Figures and Tables

**Figure 1 ijms-24-07840-f001:**
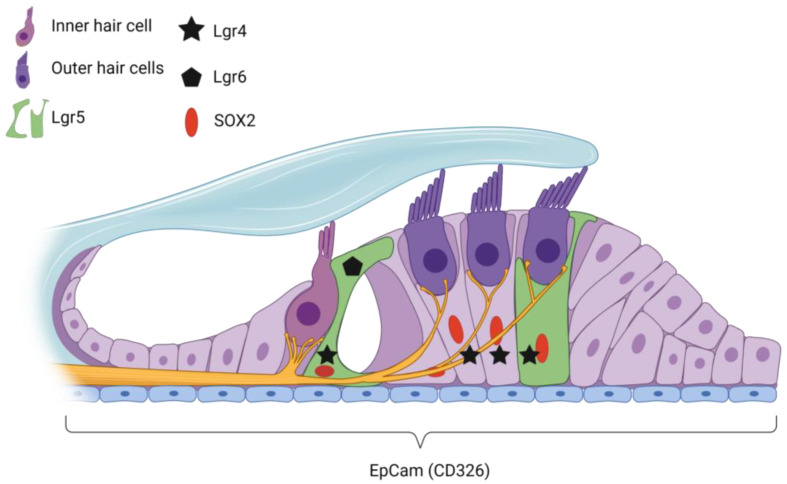
Expression of important supporting cell markers for regeneration in the adult cochlea. Lgr5-positive supporting cells (SC) indicated in green and SOX2-positive SC in red (nucleus). EpCam is expressed in all SCs in the adult cochlea. Created with BioRender.com.

**Table 1 ijms-24-07840-t001:** Markers of supporting cells with stem cell properties. Abbreviations: Border cell (BC), Deiters’ cell (DC), 3rd row of Deiters’ cell (DC3), Greater epithelial ridge (GER), Inner hair cell (IHC), Inner border cell (IBC), Inner pillar cell (IPC), Inner phalangeal cell (IPhC), Hensen cell (HeC), Lesser epithelial ridge (LER), Outer hair cell (OHC), Outer pillar cell (OPC), Reissner’s membrane (RM), Spiral ganglion neuron (SGN). Age stages (mouse) Embryonic/fetal: Before birth, Postnatal: P1–P23, Adolescent: P23+1–P60, Mature adult: >P60.

Gene/ Protein	Species	Stage	Location in Cochlea per Stage	References
EpCAM	Mouse	Mature adult	RM, Cochlear HCs, SCs	[116]
	Human	Fetal	Chochlear duct	[115,116]
Lfng	Mouse	Postnatal	DC3	[101]
		Mature adult	IphC, IPC, OPC, DC	[113]
Lgr4	Mouse	Embryonic	Cochlear duct and SGN	[23]
		Postnatal	DCs and IPCs	[23]
		Adolescent	DCs	[23]
Lgr5	Mouse	Embryonic	DC3, IPCs, IphCs, and the lateral GER	[22,82]
		Postnatal	IPC, GER, DC3, IPHC	[20,22,77,82,122]
		Adolescent	IPCs, DC3, IBC	[82,110]
		Mature adult	DC3, IPCs	[22,110]
		Deafened (p30)	Survival in DC3 only	[110]
	Human *	Fetal	Prosensory doman, LER, SCs	[115]
Lgr6	Mouse	Embryonic	IPCs	[20]
		Postnatal	IPCs, IBCs (disappears at p30)	[20,22]
Sox2	Mouse	Embryonic	HCs, adjacent supporting cells and GER	[20,120]
		Postnatal	SCs	[22,82]
		Mature adult	BCs, IphCs, IPC, OPC, DCs, HeCs	[21,121]
		Deafened (p120)	BCs, IphCs, IPC, OPC, DCs, HeCs	[21,121]
	Human	Fetal	Organ of Corti	[115]

* data from mRNA expression.

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
