# Peer review of "Regeneration of Hair Cells from Endogenous Otic Progenitors in the Adult Mammalian Cochlea: Understanding Its Origins and Future Directions"

_ijms, 2023, doi:10.3390/ijms24097840_

Round 1
Reviewer 1 Report
Smith-Cortinez et al. have comprehensively reviewed the literature regarding cochlear hair cells regeneration and IHC re-innervation. They particularly focused on inner hair cells regeneration through recently identified signalling pathways and their pharmacological modulators. They suggest that combinatorial approach, i.e. acting synergistically on that pathways, including also epigenetic modulation, may be the most relevant option to achieve regeneration in the adult mammalian cochlea. I really appreciated the reading of this review which is clear, concise, comprehensive, well written and adequately structured. I believe that most of the relevant references in the field have been cited.
Therefore, I only have very few minor comments to the authors manuscript.
Line 320: “In line with these findings, is has been”. Replace is by it.
Line 510: “5.3. Improving functional outcomes: nerve re-innervation after cochlear trauma” I would suggest either to remove “nerve” or to replace it with “IHC”
Line 530-531: I would avoid the repetitions of interestingly.
Author Response
Dear Reviewer 1,
We appreciate the time for reviewing our review article. We have made all changes suggested.
Reviewer 2 Report
In this review the authors have described various molecular signals involved in the development of the inner hair cells and strategies to regenerate hair cells and reinnervation of newly formed (regenerated) hair cells. Overall, it is a well written review. However, the authors must consider adding the following to the review.
1. The title of the review is “Regeneration of hair cells from endogenous otic progenitors in the adult mammalian cochlea: understanding its origins and future directions”. In the current format, regeneration in the adult cochlea is not elaborately described. Most of the manuscript focuses on the studies in early postnatal cochlea. Therefore, it would be better to eliminate the word “adult” from the title.
2. The authors have done a great job in summarizing the molecular signals (Notch, Wnt, Fgf, Shh) regulating the development of the inner ear from the otic progenitors. MicroRNAs are one of the key regulators of formation and maturation of hair cells (Groves et al., 2013) as well. The authors must include that in this review.
3. Some recent studies by Iyer et al., 2022 and Menendez et al., 2020 have shown combination of ATOH1, GFI1 and POU4F3 can facilitate cellular reprogramming. Although the study by Iyer et al. is in the reference list, it is not included in the text.
4. The authors might also discuss that expression pattern of some genes and/ or proteins differ in human compared to rodents and non-human primates. Therefore, the translational approach of regeneration must consider that fact as well while doing gene manipulation for reprogramming cells.
5. Hair cell regeneration strategies also require inner and outer hair cell fate specification regulation by transcription factors like Tbx2 (Bi et al., 2022; Kaiser et al., 2022), Ikzf2 (Chessum et al., 2018, Sun et al., 2021) and Insm1(Wiwatpanit et al., 2018). The role of these transcription factors must be mentioned in the development of cochlear hair cells and regeneration.
6. In line 522, please change RGM to RGMa (Nevoux et al., 2021).
Author Response
Dear Reviewer 2,
We appreciate the time and effort you took in reviewing our review article. We have answered your questions in the same order as you have written them in the revisions.
1) We understand and agree with your question regarding the use of progenitors/progenitor cells instead of stem cells. We replaced “endogenous stem cells” by “endogenous progenitor cells” throughout the manuscript. We have also included the reference by Udagawa et al. 2021. Because Kubota 2021 focused on comparing the organoid forming potential of GER cells compared to supporting cells in neonatal mice and we focus mainly on adult, we did not include this study.
2) We appreciate your suggestion to include age-related hearing loss as part of this chapter. We followed your suggestion and re-labeled the sub-chapters as age-related hearing loss (2.1), noise-induced hearing loss (2.2), and ototoxicity (2.3). Further, because cochlear synaptopathy is a consequence of noise-, ototoxic- and age-related hearing loss, we have deleted that sub-title and included the information in the relevant sections.
3) We understand your suggestion and re-labeled the title of this chapter to “cochlear development and its associated pathways” and included information on the development of spiral ganglion neuron and innervation in new Subchapter 3.2 for the reader.
4) We created a new chapter as suggested (now Chapter 4). Because in Chapter 3 we have already summarized how hair cells are produced and because our target is adult regeneration, we will focus in Chapter 4 on supporting cells with regenerative potential in the adult, as stated in Table 1. The alteration of these molecules after damage is stated later, in new Subchapter 5.3, because only a few reports have focused on these analyses.
5) Indeed, in adult cochleas, it is known that Lgr5, Lgr4, Lgr6, and SOX2 are restricted to supporting cells in the organ of Corti as described in the review. However, Yu et al. (2019) (doi: https://doi.org/10.1101/739680) described that EpCam positive cells (in adult mouse cochleas) are cells from the inner and outer sulcus regions (as cells from the Reissner’s membrane), epithelial cells from the stria vascularis, hair cells, supporting cells and melanocytes. Based on this, we have updated the figure so that it better represents the expression of EpCam in the adult cochlea.
6) We have modified the text according to your suggestions and have included the suggested literature. See Subchapter 5.1.
7) We have modified the text according to your suggestions and have included the suggested literature. See Chapter 5.2
8) The current review focuses mainly on targeting signaling pathways controlling inner ear regeneration and not transcription factor reprogramming by gene therapy. Iyer and Groves 2021 (doi: 10.3389/fncel.2021.660748) summarizes the research on that field. We have now included a line stating that in Subchapter 5.1. Regarding the paper by Matsunaga and Nakagawa 2023 (doi.org/10.3390/pharmaceutics15030777), we have included it in the new Chapter 7 about clinical trials.
9) We have now included these recent papers in our chapter on clinical trials (Chapter 7).
10) Thank you for noticing. We have corrected now the various mistakes.
Reviewer 3 Report
This is a review manuscript focusing on hair cell regeneration from supporting cells in the adult mammalian cochlea after damage. The title is attractive, but it does not reflect the contents. The authors conclude the importance of three approaches for hair cell regeneration in adult mammalian cochleae after damage: manipulation of endogenous stem cells, 2) control of chromatin accessibility, and 3) application of neurotrophins for promoting innervation in Abstract. However, the main content is the summary of previous findings for induction of hair cell (HC) regeneration by manipulation of well-known signaling pathways. The contents of chromatin accessibility and re-innervation are limited. Before publication, revision for use of technical terms and additional citation of recent studies are required.
Major critiques
1) In the title, a term of endogenous otic progenitors is used, while in the abstract, a term of endogenous stem cells is used. This should be unified. A term of ‘progenitors’ or ‘progenitor cells’ may be suitable, because the authors highlighted LGR5-expressing supporting cells (SCs) in the adult mammalian cochleae. These cells are not stem cells, although they have a potential for regeneration of hair cells. In addition, the authors should be careful for use of a term of progenitors. For instance, SCs in the avian and fish auditory epithelia have the capability for hair cell regeneration, however, they are not stem cells or progenitor cell. They are distinctly differentiated SCs.
Udagawa et al. 2021 (doi.org/10.1371/journal.pbio.3001445) demonstrated that replenishment of SCs by GER cells after ablation of Lgr5-expressing SCs, suggesting the presence of progenitor-pools in the GER. In addition, Kubota et al. 2021 (doi: 10.1016/j.celrep.2020.108646) showed the presence of principal organoid-forming progenitors in the GER. These cell populations could be stem cell-like cells in the mouse cochlear rather than LGR5-expressing SCs.
2) Chapter 2 (Hearing loss and its pathophysiology) is not a main contents, but necessary for readers, especially for non-specialists for inner ear biology. Age-related hearing loss is the predominant form of sensorineural hearing loss (SNHL). A subchapter for ageing is necessary for general introduction of SNHL and its pathophysiology.
Two subchapter describe causes and related pathology, but last one does not. If the authors focus on the pathology, titles of subchapters should be hair cell degeneration or loss and cochlear synaptopathy. Focusing the etiology for SNHL, dug-, noise-, and age-related hearing loss will be alternative choice.
3) The title of Chapter 3 is not appropriate. A term of development should be included in the title. The authors described ‘classic’ signaling pathways for hair cell development in this chapter. Initial two paragraphs should be simplified. The difference in the development of the cochlea between humans and rodents is not a matter for this manuscript. Rather than this, a brief description of innervation and related pathways should be stated, because the authors focused on re-innervation in the later parts of this manuscript.
4) Subchapter 3.5 should be separated from Chapter 3. In this subchapter, the authors describe possible markers for cochlear progenitor cells based on the literature of cochlear development. First, the authors should summarize what are expressed in distinct common progenitors of SCs and HCs during early development of the cochlea, and characterize distinct common progenitors. Afterwards, expressions of markers that the authors stated in Table 1 in distinct progenitors are to be described. In addition, alterations in expressions of these molecules after damage in adult mammalian cochleae will be important, because the authors stated that this manuscript focused on hair cell regeneration in the adult mammalian cochlea after damage.
5) The expression patterns in Figure 1 may be incorrect. Are these marker proteins negative in the inner and outer sulcus regions? Please confirm this.
6) Chapter 4 will be one of main contents. Focusing on the avian and fish auditory epithelia, in which hair cell regeneration from SCs naturally occurs, is a good idea similar to focusing on molecules and pathways that contribute to the development of cochlear HCs. Developmental processes of murine cochleae and regeneration mechanisms in the avian and fish have provided valuable suggestions for exploring novel strategies to induce HC regeneration in mature mammalian cochleae. This review manuscript lacks citations recent transcriptomic analyses of chick basilar papillae and important information of those of zebrafish lateral lines. Janesick et al. 2022 (DOI: 10.1242/dev.200113) and Matsunaga et al. 2020 (doi: 10.3389/fncel.2020.583994) demonstrated involvement of JAK-STAT signaling in the initial process of HC regeneration, which may be critical information for this review article focusing on SCs after damage. In addition, Matsunaga et al. 2023 (doi.org/10.1016/j.isci.2023.106046) reported mechanisms for SC reprogramming after damage and involvement of TGFb signaling pathways in this process in chick basilar papillae, which is also crucial information for understanding what happen in SCs after damage. In zebrafish lateral line studies, Lush et al. 2019 and Baek et al. 2022, which are cited in this review manuscript, are important. But essential information of those articles is not stated. Those studies demonstrated the presence of stem cell population in the lateral line neuromasts, which are distinctly different from cell sources of regenerated HCs. In the zebrafish lateral line, two distinct populations of stem cells and progenitors that restore HCs have been demonstrated, which is important information of those two studies.
7) The potential of neonatal mammalian cochlea for HC regeneration is also included in important contents. These will be make clear what should be manipulated in the adult mammalian cochlea.
Two articles, Liu et al. 2012 (DOI:10.1523/JNEUROSCI.0818-12.2012) and Kelley et al, 2012 (DOI:10.1523/JNEUROSCI.5420-11.2012), should be cited to state age-dependent loss of SC potential for HC regeneration. Doetzlhofer and colleagues published a series of publications stating the roles of LIN28B and follistatin in the ability of mouse SCs to HC regeneration. A recent study (Li et al. 2022 DOI: 10.1126/sciadv.abj7651) demonstrated that LIN28B and follistatin play key roles in the strict regulation of TGFb signaling, which is required for the induction of SC reprogramming in mouse cochleae. These articles should be cited.
8) Chapter 5 describes the limitation of forced expression of Atoh1 for inducing HC regeneration in mature cochleae and necessity of additional factors. These issues are well documented in previous review articles (Iyer and Groves 2021, doi: 10.3389/fncel.2021.660748; Matsunaga and Nakagawa 2023, doi.org/10.3390/pharmaceutics15030777). Citation of these articles will help simplifying subchapter 5.1. General description about pioneer factors is to be added in subchapter 5.2.
9) In Chapter 6, a part of clinical trials aiming HC regeneration are introduced briefly. Citation of recent review articles (Le Prell 2021 DOI: 10.1055/s-0041-1735522; Foster et al., 2022 DOI: 10.1002/prp2.970; Matsunaga and Nakagawa 2023, doi.org/10.3390/pharmaceutics15030777) will help understanding recent activities of clinical trials aiming HC regeneration.
10) There are many mistakes in the reference list.
Author Response
Dear Reviewer 3,
We appreciate your feedback and time.
1) In the current review, we discuss the regenerative capacity of the early postnatal cochlea, as these mechanisms can be used for the development of a potential treatment for HC regeneration in the adult cochlea. Hence, we have kept the word adult in the title.
2) We added a section on MicroRNAs (miR) in Chapter 3, Subchapter 3.2.5. This way, the chapter on development is more complete. Thank you for mentioning the absence of this part. However, in the regeneration part of our review, we will not further discuss miR3.
3) We thank the reviewer for noticing the error. Citation to Iyer has been now included in new Section 5.2 (Neonatal mammalian cochlea). The study by Menendez et al 2020 focuses on reprogramming mouse embryonic fibroblasts, adult tail-tip fibroblasts and postnatal supporting cells into induced hair cell-like cell in vitro, and for the purposes of the current review this falls outside of the scope.
4) According to a comment from Reviewer 2, we have deleted the comparisons between rodents and human development, since it was not a matter of this review. Moreover, because of the vast literature on gene manipulation (Atoh1 and other therapies), this review has not focused on gene therapy. Therefore, we included a phrase stating this in the introduction so that the reader knows early on that we will not discuss that further.
5) Although these transcription factors are indeed important for hair cell specification, our review does not further focuses on these factors so we will not include them in the development of cochlear hair cells.
6) We have changed the text accordingly.